# Image Reconstruction and Investigation of Factors Affecting the Hue and Wavelength Relation Using Different Interpolation Algorithms with Raw Data from a CMOS Sensor

Eun-Min Kim [ID], Kyung-Kwang Joo *[ID] and Hyeon-Woo Park *[ID]

Center for Precision Neutrino Research, Department of Physics, Chonnam National University, Yongbong-ro 77, Puk-gu, Gwangju 61186, Republic of Korea
* Correspondence: kkjoo@chonnam.ac.kr (K.-K.J.); 207935@jnu.ac.kr (H.-W.P.); Tel.: +82-062-530-34833 (K.-K.J.)

**Abstract:** An image processing method was employed to obtain wavelength information using light irradiated during camera exposure. Physically, hue (H) and wavelength (W) are closely related. Once the H value is known through image pixel analysis, the wavelength can be obtained. In this paper, the H-W curve was investigated from 400 to 650 nm using raw image data with a complementary metal oxide semiconductor (CMOS) sensor technology. We reconstructed the H-W curve from raw image data based on a demosaicing method with 2 × 2 pixel images. To date, no study has reported on reconstructing the H-W curve using several different interpolation algorithms in the 400~650 nm wavelength region. In addition, several factors affecting the H-W curve with a raw digital image, such as exposure time, aperture, and international organization for standardization (ISO) settings, were investigated for the first time.

**Keywords:** color space; H-W curve; digital camera; CMOS sensor; interpolation algorithms; imaging analysis



## 1. Introduction

The scintillation technique is widely used as for a target medium in neutrino detectors. Using a liquid scintillator (LS) is one method to measure light from the detector [1,2]. The LS is a mixture of a solvent and fluor, which is a scintillating powder [3]. For long-term stability in the neutrino experiments, monitoring any changes in optical or physical properties in the LS is crucial. Generally, a UV/Vis or fluorescence spectrophotometer is used to measure the absorption or emission spectra of the LS. However, such devices are expensive and require careful handling to measure optical parameters. Furthermore, these measurements require periodic extraction of a certain amount of sample from the neutrino detector, which is very cumbersome and takes time.

In the present study, we analyzed pixel images using a complementary metal oxide semiconductor (CMOS) image sensor. The feasibility of a more accessible and non-invasive method to measure an optical property from digital image data alone was investigated. Information on emission wavelengths can be obtained by taking a photo with a digital single lens reflex (DSLR) camera. When an image is taken with a camera, red, green, and blue (R, G, and B) color space values in the visible range are stored [4,5]. An image can also be represented based on the hue, saturation, and value (H, S, and V) of the color model [6,7]. Once the RGB values are known, they can be converted into HSV values. Physically, hue (H) and wavelength (W) are closely related, thus W information can be extracted from the H value [8]. The wavelength range depends on the color space, and the H-W curve can be obtained. Initially, the H-W relation was assumed to be approximately linear, but we also investigated more accurate curves in the wavelength region of interest [9–11].

Each pixel of most commercial CMOS image sensors is covered by a Bayer color filter array (CFA). Each pixel receives only a specific range of wavelengths according to the

spectral transmittance of the filter. A Bayer CFA consists of R, G, and B filters and covers a broad band of color space. In a CFA, each pixel captures just one color among R, G, or B. The other missing two-color values are estimated through a demosaicing interpolation process. There are many proposed demosaicing algorithms [12,13]. We investigated several interpolation algorithms to reconstruct the H-W curve from raw image data. In addition, some factors related to information loss during the reconstruction of the H-W curve were also investigated.

In summary, there are several motivations for this paper. For the H-W relationship, there were our previous curve measured for the wavelength of 400~650 nm [14], and CMOS or Foveon technology over 500 nm [10,11]. Because of the intrinsic limitations of the CMOS sensor we were using, the camera was not sensitive in the UV regions. Most of the signals generated in neutrino physics are read by a photo multiplier tube (PMT), and the maximum quantum efficiency of bi-alkali PMTs lies around 420 nm. The wavelength of light emitted in neutrino experiments using liquid scintillator is in the 420~430 nm region. This is the reason why we are interested in blue wavelength regions. The first was to investigate and compare H-W curves using several different interpolation algorithms in those wavelength regions. To date, there have been no such attempts. Secondly, when constructing H-W curves using raw image data, we investigated some factors affecting this curve. The most important elements, such as exposure time, aperture, and international organization for standardization (ISO) setting number, are studied. In the future, we want to develop a new method based on a digital camera to reconstruct emission wavelength from scintillators without using a spectrophotometer. Furthermore, we used a grating to obtain diffraction pattern images. Recently, gratings or profiled structures have been used in various fields like magnonic crystals [15] and magnetic impedance structures [16]. It plays an important role in data storage, transmission and processing technology by controlling the propagation and wavelength of waves. Therefore, continued and general interest to such structures is necessary.

## 2. Production of a Digital Image with CMOS Sensor Technology

### 2.1. Experimental Setup

A light emitting diode (LED) was used as a source to generate light in the entire wavelength range from 400 to 650 nm simultaneously. A Canon EOS D series (Manufacturer: Canon, Tokyo, Japan) with CFA technology was used to take digital images for those wavelength ranges. As shown in Figure 1, a single slit with a width of 0.5 mm, a collimator, and a screen were used to collect the thin light passing through the diffraction grating as much as possible. Two types of diffraction gratings, a holographic transmission diffraction grating (Manufacturer: Scientific Equipment of Houston, Houston, TX, USA) and a blazed transmission diffraction grating (Manufacturer: Edmund Optics, Barrington, NJ, USA), were tested. The holographic transmission diffraction grating has a thin surface and exhibits a sinusoidal profile, as shown in Figure 1a. The blazed transmission diffraction grating shown in Figure 1b is dug in one direction and has increased efficiency at a specific angle. We tested 830 lines/mm, and the dispersion of the light is much wider.

Many careful efforts were made to prevent external background or stray light from entering the camera lens. The position of the diffraction pattern changed depending on the wavelength of the incident light. As shown in Figure 1c, measurement results at both ends of the image appear weak, thus special care was needed. Therefore, not all areas were used for this analysis. A value (V) cut was applied to separate the diffraction grating pattern and the background. V is variable about the brightness in HSV space. High V means lighter pixel and low V means darker pixel.

After measuring the distance (Y) from the right central white image to the left edge of each color band in Figure 1d, we can obtain an angle ($\theta$) between the normal line of the ray and the diffraction image. From this, we can calculate the wavelength (W) using the Bragg formula $d\sin\theta = m\lambda$, where d is the diffraction grating slit separation, and m (an integer) is the diffraction order. If the grating has N lines per meter, then the grating spacing is

given by 1/N. As a result, the H values were obtained from the diffraction images using the distance to the white light source and the above zero-order fringe formula. Then, finally the H-W relationship can be obtained.

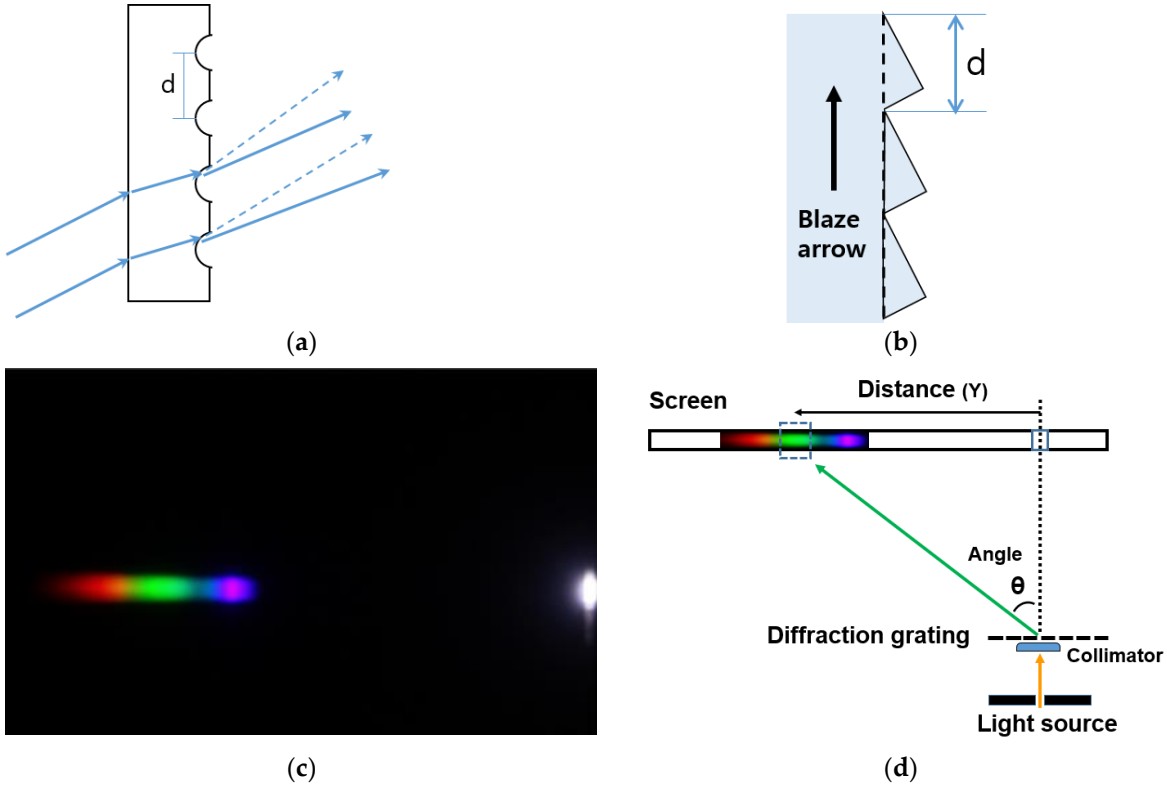

**Figure 1.** (**a**) Holographic transmission diffraction grating. (**b**) Blazed transmission diffraction grating. (**c**) A sample picture showing the photographed diffraction patterns appearing on the screen. The bright white light on the right side represents light passing through the diffraction grating. The pattern on the left side of the screen shows the image displayed on the screen after passing through the diffraction grating. (**d**) Schematics of the distance (Y) and the angle (θ) measurement between the normal line of ray and the diffraction maxima.

## 2.2. Reconstruction of Raw Image Data

### 2.2.1. Raw Image

In our study, we used a commercially available digital single lens reflex (DSLR) camera (Canon EOS D series) equipped with a CMOS image sensor. Each pixel of most commercial CMOS image sensors is covered by a CFA, and the Bayer CFA is widely used. In the CFA configuration, each pixel captures just one color among R, G, or B. The other two missing color values are estimated from the recorded mosaic data of the neighboring pixels. This interpolation process is called demosaicing and many demosaicing algorithms have been proposed [17–19].

A typical Image processing pipeline of digital camera Is shown In Figure 2. An Image pipeline is a set of components commonly used for digital image processing, consisting of several distinct processing blocks [20]. It plays a key role in digital camera systems by generating a digital color image. When an image is captured, it is initially saved as raw image data [21]. These are minimally processed data from the image sensor. The raw data files created by a digital camera contain a CFA image recorded by the photo-sensor of the camera. Each pixel of raw data is the amount of light captured by the corresponding camera photo-sensor. After storing raw data produced by a camera sensor, further processing is undertaken to generate joint photographic experts' group (jpeg) digital image.

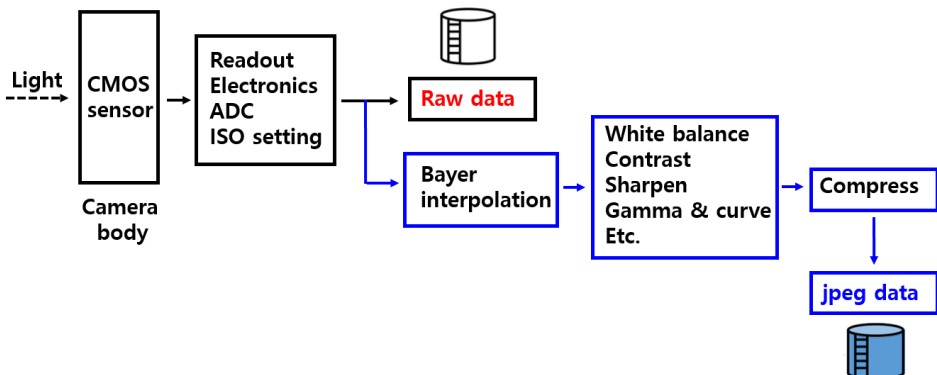

**Figure 2.** Schematic of the image pipeline for raw data and a jpeg image. Each stage of the image pipeline is fairly standard, although the stages vary in order or are combined by different digital camera manufacturers.

2.2.2. Raw Image Data Processing

The raw image is unprocessed data and is not in the form of an image. It stores light exposure data from the sensor. Raw images also contain information such as camera model, manufacturer, serial number, ISO setting number, exposure time, focal length, white balance, and color filter pattern. When light enters the camera, it passes through one of the R, G, or B filters. The intensity of the transmitted light varies depending on the wavelength of light incident on each filter. The light that passes through the filter enters the photo diode in the CMOS sensor. The incoming light creates a photocurrent that is converted to a voltage and is finally digitized. This process is performed for each pixel. This is where the camera operates and the raw image is saved.

We used the response of each pixel according to R, G, or B data for our analysis. Conventional mosaic pixel patterns captured in a single layer using the CMOS sensor CFA technology are shown in Figure 3a. In the camera we used, the pixels are configured as Bayer CFA in RG1/G2B, as shown in Figure 3b. The pixel data stored internally has a 2-D form (2 × 2 matrix array) in which light intensity data for each RGB filter is mixed. One RG1/G2B 2 × 2 array forms one unit, consisting of one R, two G (G1, G2), and one B. One photo can be considered as a 2 × 2 array arranged continuously in rows and columns. Figure 3c represents the distribution of digitized voltage values for all pixels in the raw image data.

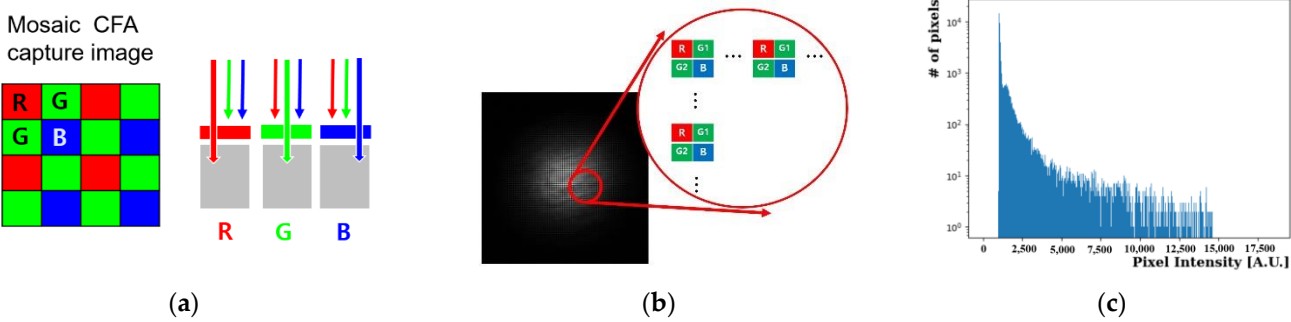

(**a**)   (**b**)   (**c**)

**Figure 3.** (**a**) Conventional mosaic pixels captured using the CMOS sensor CFA technology. Color filters are applied to a single layer of pixel sensors, and make a mosaic pattern. Only one wavelength of R, G, or B passes through a pixel and one color is recorded. (**b**) This is before the interpolation process and R, G, and B data are mixed. After processing it in black and white, each value was imaged. Pixels are internally composed of Bayer CFA in RG1/G2B 2 × 2 format. One RG1/G2B array is considered as one unit. (**c**) The digitized voltage values for all pixels. The camera we currently use digitizes at 14 bits.

To examine the R, G, and B voltage values corresponding to each pixel, photos were taken using a laser with a wavelength of 440 nm. Figure 4 shows the distribution of digitized voltages of R, G1, G2 and B, respectively. As expected, the value is largest in the B distribution and smaller in G, followed by R. After raw data were taken, RGB values are dealt through the interpolation process and converted into a 3-D color image (for example, jpeg). At this time, not only interpolation processing but also other corrections are performed, and this is called the preprocessing process. Interpolation performs demosaicing on one pixel to fill the empty filter with another color. In other words, a virtual value is given to empty data through nearest-neighbor interpolation or another algorithm. Through the preprocessing process, the RGB values are ultimately corrected to colors similar to those observed by the human eye. However, the corrected color is only a similar color and is not exactly the same as the color visible to the human eye. For this reason, various problems occur. When using jpeg images, the H value shows a significant change depending on the exposure time. This is because the RGB values change during preprocessing where color conversion, white balance, gamma correction, etc., are performed. Therefore, there was no consistent change in exposure time due to the combined effect of multiple correction processes. In addition, for the same reason as in the case of H value, the changes in S or V were inconsistent, depending on the exposure time.

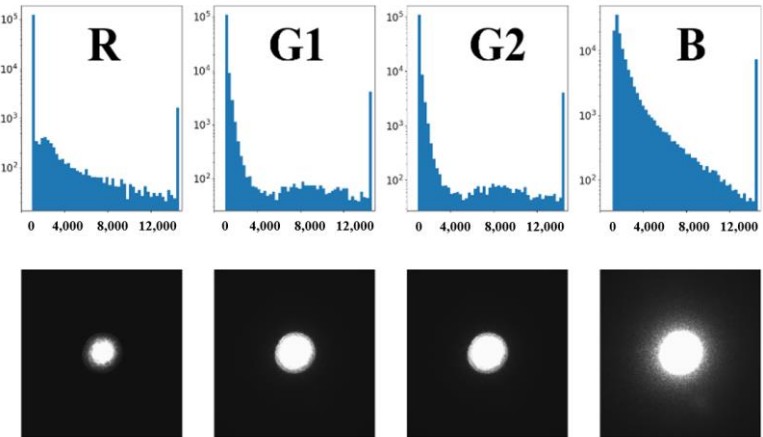

**Figure 4.** Photos corresponding to 440 nm laser illumination. From left to right, the distribution of digitized voltages of R, G1, G2 and B, respectively. Because the wavelength was 440 nm, the B component value is the largest in the distribution.

However, instead of interpolation, we performed down-sampling in this study. When down-sampling is performed, the resolution of the image itself decreases. However, since the values of surrounding pixels are used, the resolution of the H value does not change significantly, regardless of the exposure time. As the exposure time becomes shorter (i.e., a darker image), the resolution of the H distribution worsens, and in the case of images that are too dark, almost only the background can be observed. Finally, after reconstructing the RGB array through down-sampling, the color space was converted from the raw RGB color space to the HSV color space to obtain the H value.

## 3. H-W Curves with Several Different Interpolation Algorithms

### 3.1. H-W Curve Result

To obtain an H-W curve with a wide range of wavelengths, LED white light was dispersed using a diffraction grating and projected onto the screen, as shown in Figure 1. The split light was projected onto a screen and photographed with a camera. Afterward, wavelengths were distinguished according to the distance to the 1st order pattern based on the 0th order pattern, and H values were obtained for each of the 4 areas (RG1/G2B) as one unit, as explained in Figure 3b. The theoretical position of the diffraction grating pattern and the actual measured position was checked and confirmed, and there was no significant

difference between them. Figure 5 shows the mapping between W and H. There are well-known plateau regions for wavelengths in the range of 530~560 nm. At the near end of wavelengths 530 or 560 nm, the H-W response appears step-like [10,11]. These features are a direct result of the CFA color filters used in the CMOS sensor. Around ~560 nm, only the green component exists. Because neither the blue nor red filters transmit significantly in this region, the plateau naturally occurs. On the other hand, some uncertainty due to the beam size and exposure time cannot be avoided. Several intensity-related variables were investigated; these will be discussed in detail in Section 4.

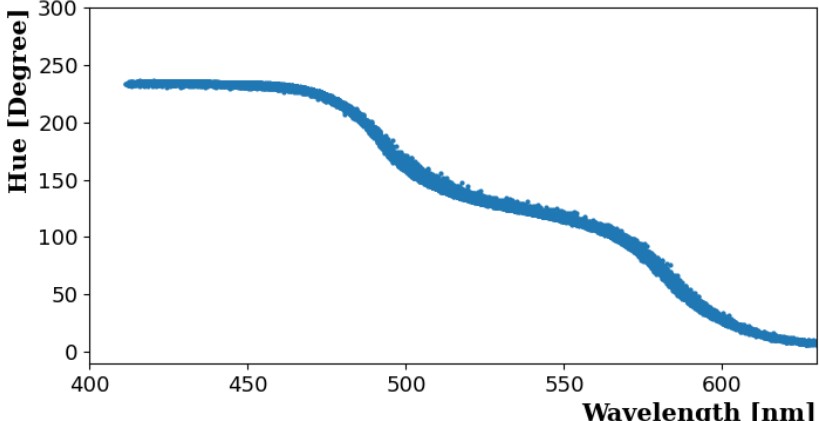

**Figure 5.** H-W curve with CMOS using CFA technology for the wavelengths from 400 to 650 nm. The raw image was used for analysis with grating (830 lines/mm). Our camera did not respond well below ~400 nm. To date, the H-W relation has not been well measured below ~500 nm. The plateau region occurs for wavelengths between 530 and ~560 nm.

The thickness does not show the average value of the selected area in Figure 1c,d, but the distribution of all points after V cut in that selected area. No matter how poorly we measure, the H-W results lie within this band.

### 3.2. Several Interpolation Algorithms: Nearest Neighbor, Linear, and Cubic

Interpolation is a process of estimating the value of a pixel that does not receive any information using known values surrounding pixels. In the case of the nearest neighbor interpolation shown in Figure 6a, a pixel value close to the empty pixel value of the CFA filter pattern in Figure 3a was inserted. It is a basic interpolation that puts four times as many pixels on the picture as are present in the image. For linear interpolation, a straight line is drawn between the two points, and that straight line is used to estimate f(x), as shown in Figure 6b. Linear interpolation is quick and easy, but it is not very precise. Figure 6c represents a cubic case, in which a cubic function is assumed between two points and that function is used to estimate f(x).

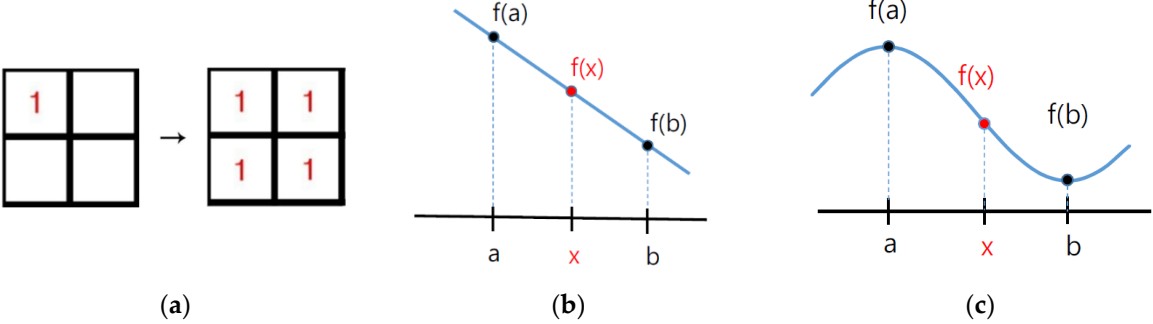

**Figure 6.** Several interpolation algorithms. (**a**) Nearest neighbor; (**b**) Linear; (**c**) Cubic.

Furthermore, there are "Area" and "Lanzcos4" interpolation algorithms. Area interpolation is mainly used to reduce images. Empty pixels are determined by averaging the surrounding pixel values. Lanczos4 is an interpolation method using the 4th order of the Lanczos function [=sinc(x)*sinc(x/a), where sinc(x) = sin($\pi$x)/($\pi$x)]. The results obtained by applying various interpolation methods and their differences are shown in Figure 7. Interpolation can be used for the purpose of reconstructing a sharper or clear image. Interpolated pixels are constructed using information from adjacent known pixels; the accuracy of H-W curve would not be significantly affected because the more actual H information in each pixel is not obtained. Therefore, there are no big differences in the H-W curve between each interpolation algorithm.

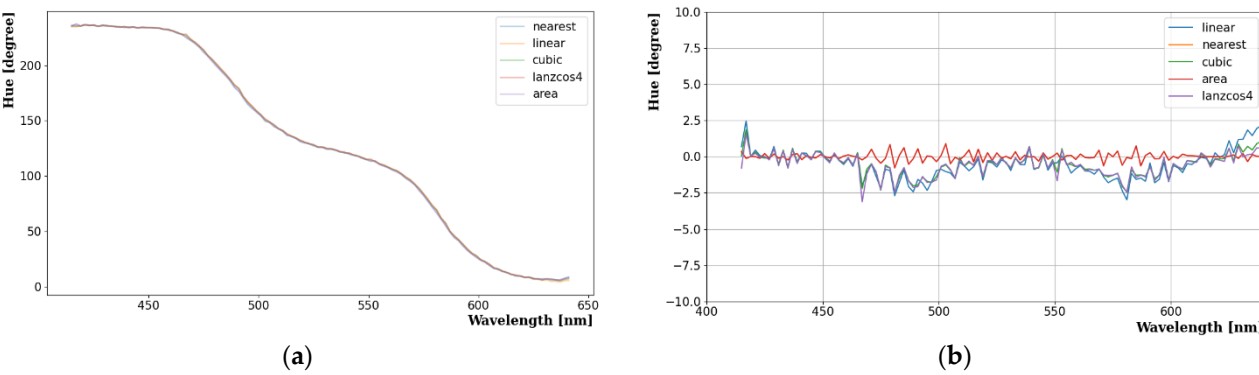

(**a**) (**b**)

**Figure 7.** (**a**) H-W curves with several interpolation algorithms. (**b**) The difference between the H-W curve value in Figure 6 and the value measured by each interpolation algorithm.

## 4. Several Intensity Variables Affecting Raw Images

The three most important components of photographic exposure that the user can adjust when taking a photo are exposure time (or shutter speed), aperture, and ISO number. We investigated how these affect the raw image results and what differences they ultimately make to the H-W curve.

### 4.1. Exposure Time (or Camera Shutter Speed)

Among the various variables that affect the results of raw images, we compared the results by adjusting the variables related to luminosity. This is because the variables related to luminosity only affect overall brightness, so we expected that there would be no correlation with H as they do not affect saturation. When using raw images, the maximum amount of charge in each pixel is set. Therefore, when light exceeds a certain intensity, saturation occurs and dumping occurs at one value. In the camera used, the maximum charge value was $2^{14}$ for each pixel. As the exposure time was increased, the brightness became greater, and the number of saturated pixels increased. The time at which light enters the camera lens was adjusted to prevent saturation. H-W curves with different exposure times from 0.1 to 20 s are drawn in Figure 8. Since the exposure time of 20 s is an extreme case, it somewhat deviates when compared to other exposure time results with large error bars. Twenty seconds already contains saturated information. Other than this, there is no significant change in H value.

### 4.2. Aperture

An aperture is a hole through which light travels. The amount of light entering the camera lens was adjusted to prevent saturation. Aperture, often called F-stop, controls the exposure of a photograph, but it also affects how much of the image is in focus, or the depth of field. The F number represents aperture light. This number controls the amount of incoming light. A large F number means that the diameter of the aperture is set very tight. Regardless of the aperture varying from F/2.8 to F/11, the average H values were not significantly different, as shown in Figure 9.

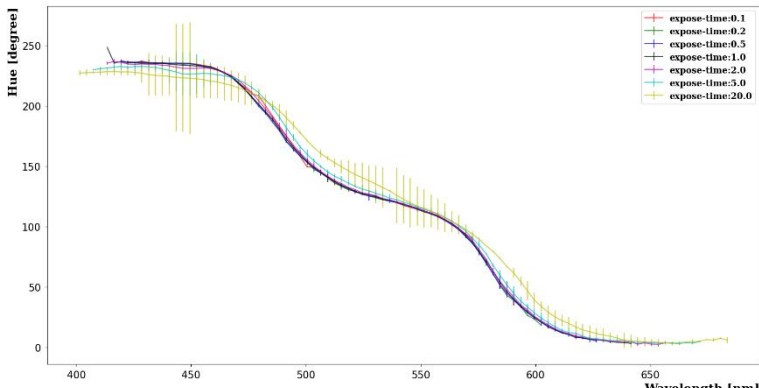

**Figure 8.** H-W curves with different exposure times from 0.1 to 20 s. Exposure time of 20 s is an extreme case.

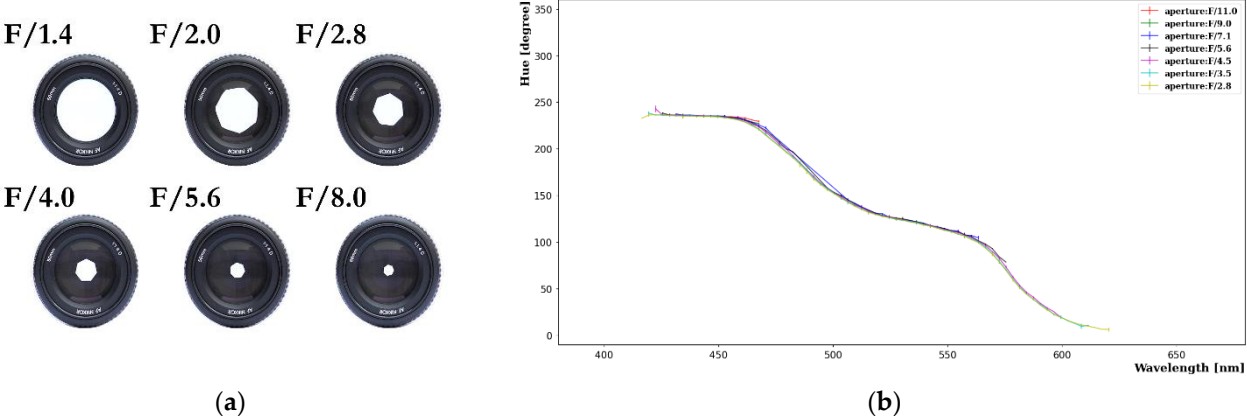

(**a**)                                         (**b**)

**Figure 9.** (**a**) Different apertures of a lens (for example, F/1.4, F/2, F/2.8, F/4, F/5.6, F/8). For each step up, the amount of light decreases by 1/2. (**b**) H-W curves with different aperture numbers.

### 4.3. ISO Number

Imaging sensor signal is amplified and digitized. The ISO number refers to the degree of amplification when digitizing an analog signal [22]. As shown in Figure 10, amplification assists analog signal to digital conversion. The higher the ISO number, the brighter the picture. ISO 100 means no amplification, and ISO 1600 means 16 times the amplification. The ISO value indicates a camera sensor's sensitivity to light. By adjusting the ISO number, even in low light or dark places, we can take pictures with some brightness. However, a higher ISO number leads to an equally higher noise as well and makes the sensor more likely to saturate.

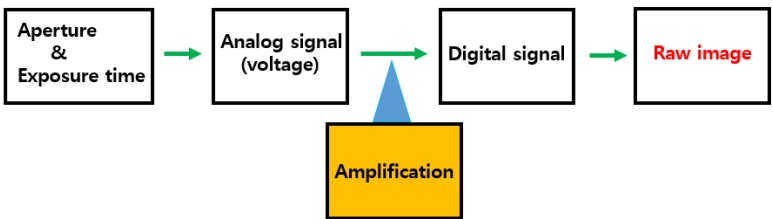

**Figure 10.** ISO pipeline in the digital camera. ISO represents the degree of amplification. Amplification assists analog signal to digital conversion.

For the pixels that had no saturated area and exhibited a certain brightness, the corresponding average H value for the entire pixel did not change significantly in the wavelength region from 400 to 650 nm, as shown in Figure 11. If the ISO is too small, the

signal cannot be clearly distinguished from the background. Conversely, if the ISO is too high, the saturation area increases because of increased brightness. The fluctuations in the wavelength regions below 400 nm or over 650 nm are due to an increase in noise due to an increase in amplification rate. The relevant pixels have only one color of R, G, or B.

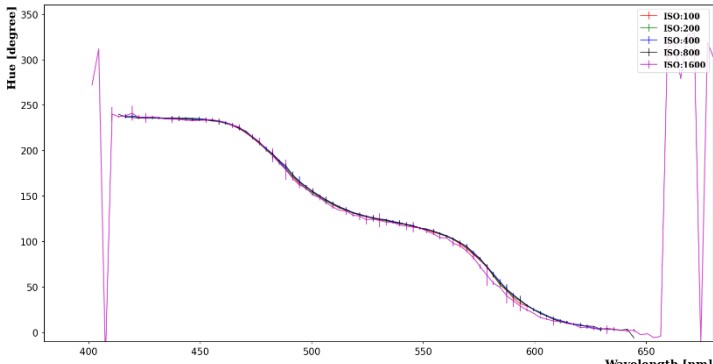

**Figure 11.** H-W curves with different ISO numbers. The huge fluctuations in the wavelength regions below 400 nm or over 650 nm are an example of high ISO 1600. At both extremes of wavelength, the amount of change is noticeable because the influence of noise is large.

## 5. Summary and Discussion

The H-W curve from 400 to 650 nm was investigated using a digital camera based on the CMOS sensor technology. We focused on these wavelength regions, since most of the fluors used in neutrino physics emit light in this wavelength range. Raw image data based on a down-sampling method with $2 \times 2$ pixel images were used to reconstruct the H-W curve. In addition, for the first time, several different interpolation algorithms were used to obtain H-W curves. If the RGB value of each pixel is known, then the HSV value can be obtained from this known RGB information. Subsequently, the H value can be converted to wavelength, when the information contained in the HSV color space is converted to a visible light spectrum. In addition, some factors affecting the H-W response were investigated. In particular, factors related to intensity, such as exposure time, aperture, and ISO number, were studied. Checking for variations in the H-W curve for the three variable changes confirmed that there was no significant change in the curve by applying appropriate selection criteria. That is, there was no drastic change in the H values, in the area where there was no saturation.

Finally, by analyzing the images taken by the digital camera, it could be possible to estimate the fluor emission spectrum in the visible wavelength region [23]. In addition, the possibility of finding the fluor component dissolved in the LS can be investigated. This will provide an inexpensive, non-invasive method of fluor emission spectrum or component analysis in a sealed LS detector. We hope that this method can potentially replace the conventional spectrophotometry techniques in the future.

**Author Contributions:** Conceptualization, K.-K.J.; methodology, H.-W.P.; software, E.-M.K.; validation, H.-W.P.; formal analysis, E.-M.K.; investigation, K.-K.J.; resources, H.-W.P.; data curation, E.-M.K. and H.-W.P.; writing—original draft preparation, K.-K.J.; writing—review and editing, K.-K.J. and H.-W.P.; visualization, E.-M.K.; supervision, K.-K.J. All authors have read and agreed to the published version of the manuscript.

**Funding:** This research received no external funding.

**Informed Consent Statement:** Not applicable.

**Data Availability Statement:** Not applicable.

**Acknowledgments:** This work was supported by grants from the National Research Foundation (NRF) of the Korean government (2022R1A2C1006069, 2022R1A5A1030700).

**Conflicts of Interest:** The authors declare no conflict of interest.

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
