# Peer review of "Image Reconstruction and Investigation of Factors Affecting the Hue and Wavelength Relation Using Different Interpolation Algorithms with Raw Data from a CMOS Sensor"

_photonics, doi:10.3390/photonics10111216_

Round 1

Reviewer 1 Report

Comments and Suggestions for Authors

Submitted manuscript is interesting and it might be the subject of publication after revision. The first point to address is the novelty of research. Recently authors had published the paper entitled “Investigation of the Hue–Wavelength Response of a CMOS RGB-Based Image Sensor”, it looks like two works have points of similarity. Authors must refer to the published work and outline the need of additional research.

At the end of the Introduction a clear goal of the present study must be formulated without long description of the previous works and references – clear and short aim of the study. Gratings or profiled strictures were recently used in many areas like magnonic crystals, BFG, magnetic impedance structures, it would be nice to mention the existence of general interest to such structures.

Figure 3 must be given as a supplementary material.

It is necessary to provide estimation of the experimental errors for the figures like 8 or 9 may be with reference on the aperture features (Fig. 10).

The last part of the Summary belongs to the Discussion part, it should be placed accordingly.

Comments on the Quality of English Language

Reasonable

Author Response

Please take a look at an attached file. Thanks. 

Reviewer 2 Report

Comments and Suggestions for Authors

1. In section 2.2, reconstruction of raw image date is the core part of this work, and the authors could have written a little more detailed in this aspect, such as adding some theoretical derivation and analysis.

2. In order to highlight the applicability of this work, the authors could add some description in the introduction section.

3. Some figures are not clear enough. For example, Figs. 3, 5, 8 and 12. Please improve.

4. The authors could reference more previous related work. Also, the authors could make comparisons with a sufficient number of relevant papers that are among the most recently published. The advantage and new contribution of this work should be further justified. 

Author Response

(The authors gave the same response as above.)

Reviewer 3 Report

Comments and Suggestions for Authors

There is a lack of originality/novelty for the content discussed in figures 1,2,3,4. While I appreciate the authors' effort to explain digital image processing, the related content can be summarized by citing appropriate literature, which is largely available from numerous sources.

The authors need to improve further on explaining their results. For example, the authors used several interpolation algorithms to model H-W curves. However, the results presented in Figure 8 show that there is no significant difference among these algorithms.  Could the authors explain the results in Figure 8 and present to the reader how we should interpret such data and modeling curves? 

Overall, I think the manuscript needs improvements to trim down the background information part while focusing on explaining what's new and novel in the results section.

Author Response

(The authors gave the same response as above.)

Round 2

Reviewer 2 Report

Comments and Suggestions for Authors

Thanks to the authors for the revisions. My concerns have been addressed.

Reviewer 3 Report

Comments and Suggestions for Authors

The authors addressed my questions from the previous review in the current manuscript. I have no other questions.